# Widespread extracellular electron transfer pathways for charging microbial cytochrome OmcS nanowires via periplasmic cytochromes PpcABCDE

Pilar C. Portela [1,2,3,4,5], Catharine C. Shipps[1,2,5], Cong Shen [1,2], Vishok Srikanth [1,2], Carlos A. Salgueiro [3,4] ✉ & Nikhil S. Malvankar [1,2] ✉

Extracellular electron transfer (EET) via microbial nanowires drives globally-important environmental processes and biotechnological applications for bioenergy, bioremediation, and bioelectronics. Due to highly-redundant and complex EET pathways, it is unclear how microbes wire electrons rapidly ($>10^6 s^{-1}$) from the inner-membrane through outer-surface nanowires directly to an external environment despite a crowded periplasm and slow ($<10^5 s^{-1}$) electron diffusion among periplasmic cytochromes. Here, we show that *Geobacter sulfurreducens* periplasmic cytochromes PpcABCDE inject electrons directly into OmcS nanowires by binding transiently with differing efficiencies, with the least-abundant cytochrome (PpcC) showing the highest efficiency. Remarkably, this defined nanowire-charging pathway is evolutionarily conserved in phylogenetically-diverse bacteria capable of EET. OmcS heme reduction potentials are within 200 mV of each other, with a midpoint 82 mV-higher than reported previously. This could explain efficient EET over micrometres at ultrafast ($<200$ fs) rates with negligible energy loss. Engineering this minimal nanowire-charging pathway may yield microbial chassis with improved performance.

Common soil and marine microbes of the family *Geobacteraceae* are important in diverse natural environments and for biotechnological applications[1]. To survive in the absence of oxygen or other soluble electron acceptors, these microbes export (donate) electrons produced during respiration to external electron acceptors, such as insoluble metal oxides (iron, uranium, etc.) and electrodes, using a process called extracellular electron transfer (EET), or to syntrophic partners in a process called direct interspecies electron transfer (DIET)[1,2]. EET[3] and DIET[4] are essential for these microbes to survive in

extreme environments by forming a community. However, in many settings, EET and DIET via soluble mediators become inefficient due to diffusion and flow forces that are typical of microbial environments[5].

To overcome this problem, *Geobacter*[6] and other species[7,8] use conductive filaments, called nanowires, capable of fast, long-range EET (Fig. 1a). Although filamentous appendages involved in EET were first reported over two decades ago[9], until very recently these appendages were thought to be Type-4 pili[6,10]. *G. sulfurreducens* pili structure and localization are akin to Type-2 secretion endopili, suggesting that pili

[1]Microbial Sciences Institute, Yale University, West Haven, CT, USA. [2]Department of Molecular Biophysics and Biochemistry, Yale University, New Haven, CT, USA. [3]Associate Laboratory i4HB – Institute for Health and Bioeconomy, NOVA School of Science and Technology, Universidade NOVA de Lisboa, Caparica, Portugal. [4]UCIBIO – Applied Molecular Biosciences Unit, Department of Chemistry, NOVA School of Science and Technology, Universidade NOVA de Lisboa, Caparica, Portugal. [5]These authors contributed equally: Pilar C. Portela, Catharine C. Shipps. ✉e-mail: csalgueiro@fct.unl.pt; nikhil.malvankar@yale.edu

**Fig. 1 | Purification and characterization of cytochrome OmcS nanowires. a** TEM image of *G. sulfurreducens* Δ*omcZ* cell producing OmcS nanowires. **b** OmcS structure showing stacked hemes (PDB ID: 6EF8) **c** Size-exclusion chromatogram of OmcS nanowires. **d** TEM of purified OmcS nanowires. **e** Coomassie-stained SDS-PAGE gel of purified OmcS nanowires showing sample purity. The full-length gel is shown in Supplementary Fig. 1. All experiments were performed at least in triplicate and yielded similar results.

serve as pistons to secrete cytochromes that polymerize into nanowires[10]. *G. sulfurreducens* requires nanowires of cytochrome OmcS[11] to eliminate respiratory electrons via EET to soil-abundant Fe(III) oxide and DIET[6], and OmcZ nanowires to grow on electrodes and generate electricity[8,12], whereas OmcE was found to polymerize under bacterial growth conditions that do not require EET or DIET[13]. Structures of these nanowires reveal interconnected chains of cytochromes encasing stacked heme cofactors arranged in parallel (3.4–4.1 Å) and T-stacked (5.4–6.1 Å) sequential pairs[11] (Fig. 1b). Such closely stacked hemes can promote rapid and insulated electron conduction over distances of several micrometers[8,11,12].

Although nanowires transport electrons received from within the cell to external acceptors, there are large spatial, kinetic and thermodynamic barriers to move electrons from the inner-membrane to the outer-surface nanowires[14]. The periplasmic compartment in *G. sulfurreducens* is significantly larger (>40 nm) than that of most microbes (<30 nm)[15], which is too large for direct electron transfer limited to -1.5 nm[16]. In addition to this large spatial barrier, cells pass electrons rapidly[17] (>$10^6$ s$^{-1}$) to outer-surface nanowires despite the kinetic barriers of a crowded periplasmic environment[18] and slow (<$10^5$ s$^{-1}$) electron diffusion within monomeric periplasmic cytochromes[19,20], which will require at least 10-times higher abundance of periplasmic cytochromes than nanowires to match the rate. As OmcS nanowires typically contain 200 monomers, this will require at least more than 2000 monomers of periplasmic cytochromes to overcome the diffusion limitation assuming the electron transfer rate is similar among OmcS

and periplasmic cytochromes. There also appears to be a large (-100 mV) thermodynamic barrier for electron transfer from periplasm to nanowires because the reduction potential of the periplasmic cytochromes (Table 1)[21] is -100 mV more positive compared to OmcS (−212 mV)[22] (all potentials are reported vs SHE), although in this study OmcS was not native but rather purified with detergent and boiled in a denaturant[22], which could affect the reduction potential. As a result, models of EET proposed that periplasmic cytochromes do not transfer electrons directly to OmcS and instead require several intermediate cytochromes, such as OmcB and OmcE[23]. Furthermore, the substantially negative reduction potential of OmcS is noted to be the opposite of how electron transfer chains are commonly arranged[24]. Besides outer-surface nanowires, *G. sulfurreducens* also relies on five porin-cytochrome complexes on the outer-membrane, among which ExtABCD are essential in bacterial growth only on electrodes, which do not require OmcS nanowires[25,26].

Here, we perform a suite of biochemical, electrochemical, and spectroscopic measurements on intact OmcS nanowires under physiologically relevant conditions. We find a reduction potential of −130 mV (Fig. 2e), which is 82 mV more positive than that reported previously[22], thus eliminating the thermodynamic barrier. We find that all major periplasmic triheme cytochromes PpcABCDE (PpcA-E) transiently bind to hexaheme OmcS nanowires to transfer electrons, thus eliminating the spatial and kinetic barrier. This finding contrasts with the widely used concept that EET requires periplasmic cytochromes to exchange electrons among each other[23,27,28]. This defined nanowire-charging pathway is widespread in environmentally important bacteria capable of performing EET and DIET.

## Results and discussion

### Purification and characterization of OmcS nanowires

We purified OmcS nanowires from *G. sulfurreducens* strain Δ*omcZ* to avoid any contamination from OmcZ nanowires[29] (Fig. 1a). We sheared the nanowires from cells via blending and purified them using size-exclusion chromatography (Fig. 1c) (see Methods for details). Both transmission electron microscopy (TEM, Fig. 1d) and SDS-PAGE Gel of purified filaments (Fig. 1e) confirmed sample purity suitable for all experiments described in this work.

### Spectroelectrochemistry reveals physiologically relevant reduction potential for OmcS nanowires

To determine the physiologically relevant reduction potential of OmcS nanowires with high accuracy, we performed spectroelectrochemistry[30] (SEC) at pH 7 (see Methods for details) (Fig. 2a, b, Supplementary Fig. 2). Surprisingly, we found an apparent midpoint reduction potential $E_{app} = −130 ± 13$ mV (Fig. 2c), which is 82 mV more positive than the previously reported potential for non-native OmcS[22]. The redox-active range for OmcS nanowires was 360 mV (−0.31 to 0.05 V), 40 mV larger than the prior study[22]. Thus, our measurements of intact OmcS nanowires yielded a reduction potential comparable to that of periplasmic

## Table 1 | Apparent midpoint reduction potential ($E_{app}$), redox-active window and individual heme reduction potential values of periplasmic cytochromes PpcA-E at pH 7

| Protein | $E_{app}$ (mV) | Redox-active window (mV) | Reduction potential for each heme (mV)[71] | Expression level (reads per kilobase mapped) for mid-exponential cells growing with fumarate[36] |
|---|---|---|---|---|
| PpcA[67] | −117 | 285 | (I) − 147; (III) − 104; (IV) − 110 | 7371 |
| PpcB[67] | −137 | 270 | (I) − 146; (III) − 155; (IV) − 119 | 66 |
| PpcC[72] | −143 | 265 | ND | 37 |
| PpcD[72] | −132 | 275 | (I) − 149; (III) − 96; (IV) − 151 | 813 |
| PpcE[72] | −134 | 280 | (I) − 154; (III) − 160; (IV) − 96 | 109 |
| Non-native **OmcS**[22] | −212 | 320 | ND | ND |
| **OmcS Nanowire** (This work) | −130 | 360 | ND | ND |

These values are represented in Fig. 8a.
*ND* not determined.

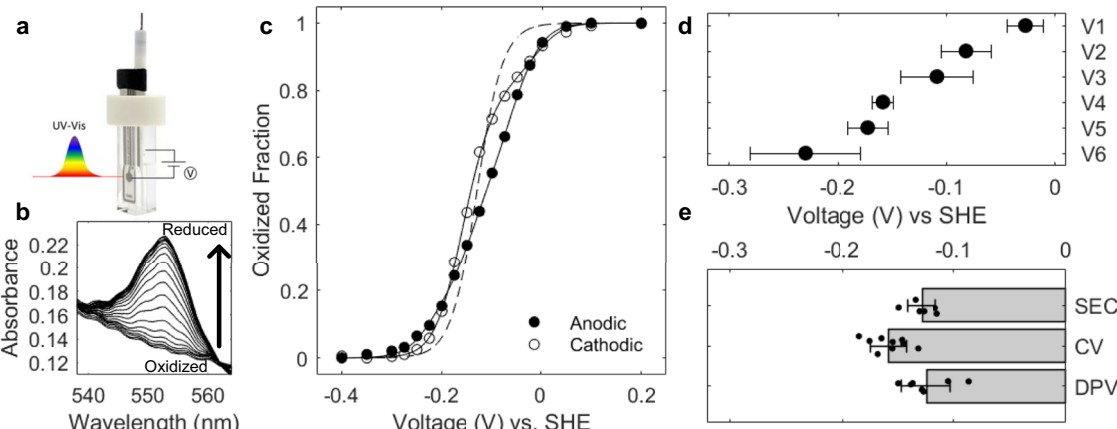

**Fig. 2 | OmcS heme reduction potentials are within 200 mV of each other, with a midpoint 82 mV higher than reported previously. a** Spectroelectrochemistry (SEC) setup. **b** Representative changes in the UV-visible α-peak spectra of OmcS nanowires upon reduction. **c** Change in the oxidized fraction of OmcS nanowires, calculated from the area of the α-peak in (**b**). Six-heme independent Nernst curves fitted to representative anodic and cathodic traces (solid lines) and compared to a theoretical Nernst with a reduction potential of −130 mV for all hemes (dashed line). **d** Reduction potentials of six hemes (V1–V6) calculated from the solid fit lines in (**c**) by averaging the anodic and cathodic traces from three replicates (*n* = 6). **e** Reduction potential of OmcS nanowires measured in solution vs. solid-state are comparable (*n* = 6,9,7 for SEC, CV cyclic voltammetry, and DPV differential pulse voltammetry, respectively). Data were presented as mean values ± s.d for (**d**, **e**).

cytochromes and a 40 mV larger redox-active window (Table 1). At the time of the prior study[22], it was not known that OmcS could form nanowires. As a result, the study subjected OmcS to detergents and boiling in the denaturant sodium dodecyl sulfate which may have broken the nanowires and thus left the hemes more solvent exposed. Solvent exposure can stabilize the oxidized state of hemes, thereby making the redox potential more negative[31]. The reduction potential of solvent-exposed hemes in microperoxidases (−0.190 to −0.220 V)[31] is similar to non-native OmcS[22]. Notably, the circular dichroism spectra of non-native OmcS[22] lacked the cotton effect (crossover from positive to negative ellipticity), which is a hallmark of excitonic coupling in OmcS nanowire hemes[8], further confirming that prior studies did not use intact OmcS nanowires for the measurement of reduction potential.

### Heme reduction potentials are within 200 mV of each other

Remarkably, all six hemes in OmcS showed reduction potential within ~200 mV (Fig. 2d). This energy landscape could explain the remarkable ability of OmcS nanowires to transport electrons over micrometer distances with negligible energy loss[16], at ultrafast (<200 fs) rates[29]. Monomeric multiheme cytochromes (e.g. MtrC that forms a porin-cytochrome complex) typically show >300 mV difference between heme redox potentials[32], which could explain their lower electronic coupling[16], and >30-fold slower electron transfer than OmcS nanowires[33]. Thus, we find that the unique heme architecture in OmcS nanowires can move electrons efficiently over micrometers.

### Surface-adsorbed OmcS nanowires show reduction potential similar to solution measurements

As microbial nanowires need to bind to solid electron acceptors for EET[34], we measured the reduction potential of OmcS nanowires adsorbed on gold electrodes[35]. Atomic force microscopy and infrared nanospectroscopy imaging have shown that OmcS nanowires adsorbed on gold electrodes show an overall secondary structure similar to cryo-EM structure and cover the entire surface without specific orientation[12]. Complementary differential pulse voltammetry (DPV) and cyclic voltammetry (CV) measurements confirmed that the midpoint reduction potential of OmcS nanowires does not change significantly in solution vs. solid-state. However, individual heme reduction potentials could vary (Figs. 2d, e, 3). This robust midpoint reduction potential of OmcS nanowires could help bacteria in EET to both solid surfaces, such as minerals in the soil and electrodes, and in DIET to syntrophic partners in co-cultures in solution with high efficiency without any energy loss.

### NMR shows PpcA-E can individually inject electrons into OmcS nanowires, but at differing efficiency with the least-abundant PpcC showing the highest electron transfer efficiency

Genetic studies have shown that periplasmic cytochromes PpcA-E are involved in EET[36,37]. Still, the possibility that they could be passing electrons to nanowire-forming OmcS has been overlooked because of extremely negative reduction potential (−212 mV) of non-native OmcS[22] compared to the most abundant periplasmic cytochrome PpcA (−117 mV)[21] (Table 1). Therefore, it has long been believed that periplasmic carriers must transfer electrons among themselves before passing to outer-surface filaments[23]. However, our finding of the reduction potential of OmcS nanowires being comparable to that of periplasmic carriers suggested that periplasmic cytochromes could be directly injecting electrons into OmcS nanowires. Here, we use the term electron injection to highlight the specifically targeted electron flow between a low potential molecule to a high potential one[38].

To directly evaluate this possibility, we measured electron transfer between PpcA-E and OmcS nanowires using an NMR-based method we developed[39]. We used PpcA-E (in the reduced state) and OmcS nanowires (in the oxidized state) (Fig. 4a). The OmcS monomer is ca. 5 nm long and has a molecular weight of ~50 kDa[11]. Therefore, a nanowire of a typical length of 1 μm (Fig. 1a) contains at least 200 monomers with a total molecular weight of 10,000 kDa (50 kDa × 200). The molecular weights of PpcA-E[21] are ~10 kDa, which are 1000-fold lower than OmcS nanowires, causing the NMR signals of PpcA-E to be narrow and easily distinguishable in contrast to those from the OmcS nanowire (Fig. 4b). In addition, in the reduced state, PpcA-E signals cover the −2 to 11 ppm range, whereas in the oxidized state the signals spread over a larger spectral window (−5 to 27 ppm) due to the paramagnetic effect of the unpaired electron of each heme iron (Fig. 6). By monitoring the spectral changes in NMR signals of PpcA-E, we visualized the electron transfer from each of PpcA-E to OmcS nanowires (Figs. 4b, 5).

As an OmcS monomer contains six hemes and each PpcA-E contains three hemes, the addition of oxidized OmcS nanowires to reduced PpcABDE in a 1:1 heme ratio maintained their reduced state (Figs. 4b, 5). Let's consider that at least one heme in OmcS can receive electrons from PpcA-E, at a 1:1 heme ratio, for each OmcS molecule. We have two molecules of PpcA-E and only one PpcA-E molecule will be

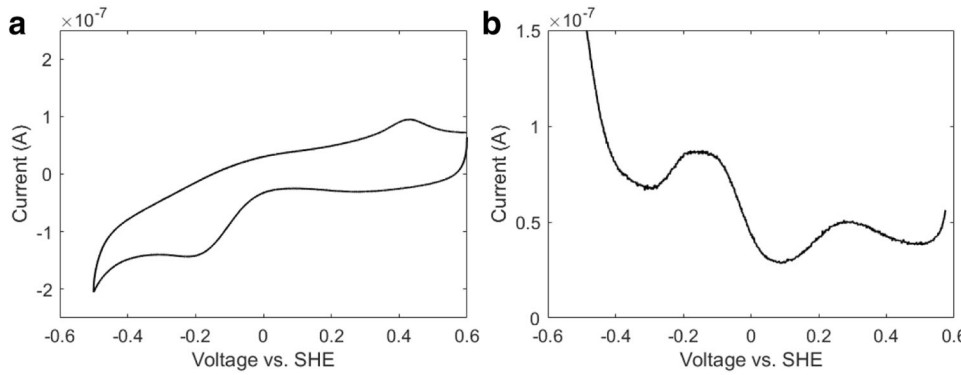

**Fig. 3 | Representative voltammetric curves. a** A representative curve of Cyclic Voltammetry and **b** Differential Pulse Voltammetry of OmcS nanowire films on gold electrodes.

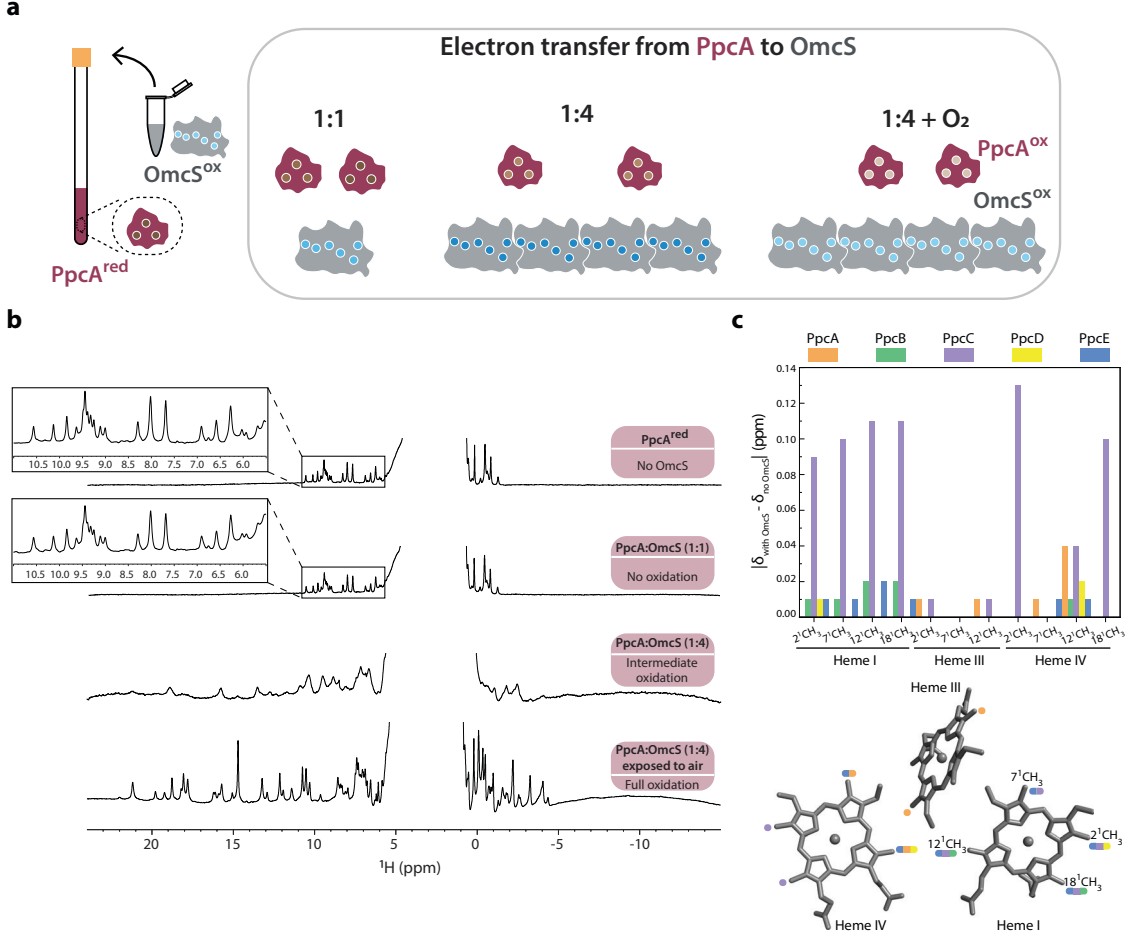

**Fig. 4 | PpcA injects electrons into OmcS nanowires with differing efficiency, with PpcC showing the largest interaction with OmcS. a** Strategy to measure PpcA (red) injecting electrons into OmcS (gray) with corresponding hemes in reduced (dark brown or dark blue) and oxidized (light brown or light blue) states.

**b** Representative NMR spectra showing PpcA oxidation. **c** Change in the chemical shift of color-coded PpcA-E heme methyls upon binding to OmcS nanowires (PpcA – orange; PpcB – green; PpcC – purple; PpcD – yellow; PpcE – blue). The most affected heme substituents are indicated in PpcA's heme core (PDB ID: 2MZ9).

able to transfer electrons to OmcS, thus existing excess of reduced PpcA-E molecules. Only when we added an excess of OmcS to reduced PpcABDE (4:1 heme ratio) the NMR spectrum showed oxidized signals in regions 10 to 27 ppm and −5 to 0 ppm (Figs. 4b, 5), as demonstrated by their similarity to air-oxidized PpcA at various stages of oxidation (Fig. 6). At a 4:1 heme ratio, there will be excess of OmcS molecules in relation to PpcA-E and, consequently, electrons will be transferred from PpcA-E to OmcS. Thus, PpcABDE readily transferred electrons to OmcS nanowires. As this electron transfer was incomplete,

we found a mixture of reduced and oxidized PpcABDE in the solution, as expected. This mixed redox state suggests a bacterial strategy to smoothly control the electron transfer into nanowires to modulate EET and DIET rates without causing a bottleneck by completely reducing or oxidizing the cytochromes.

Surprisingly, PpcC showed slight oxidation even in the 1:1 heme ratio, indicating higher electron transfer efficiency to OmcS than the other periplasmic cytochromes (Fig. 5b). This efficient oxidation is due to higher binding to OmcS, as quantified by NMR chemical shift

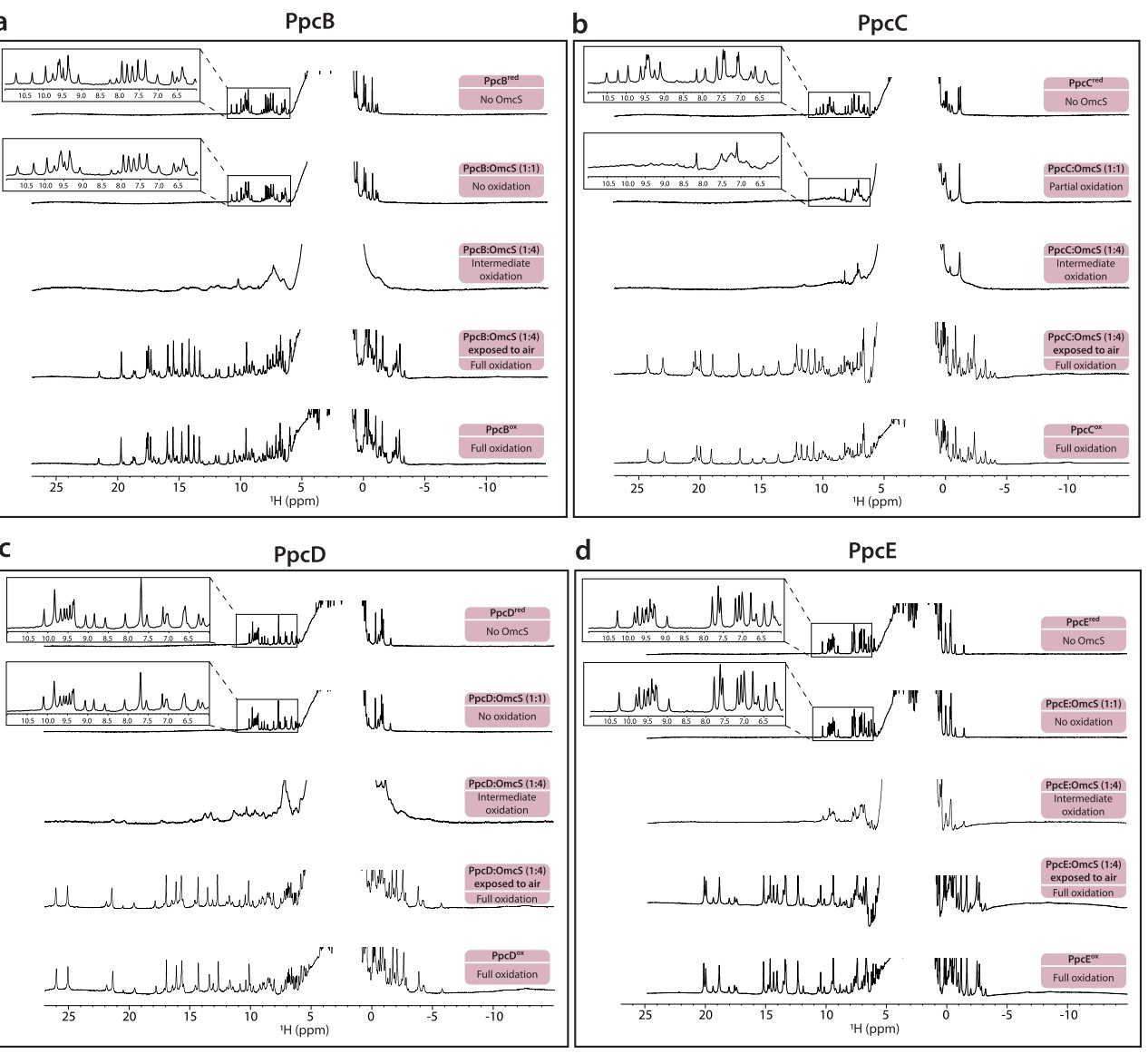

**Fig. 5 | PpcB-E charge OmcS nanowires at differing efficiency, with PpcC showing the highest efficiency.** Representative NMR spectra showing five stages of oxidation for **a**, PpcB, **b**, PpcC, **c**, PpcD, and **d**, PpcE.

perturbation experiments (Fig. 4c). As PpcC is the least-abundant cytochrome in the periplasm, our finding of the highest electron transfer efficiency of PpcC could explain the ability of *G. sulfurreducens* to continue EET even only in the presence of one of the PpcA-E paralogs[31].

### Electron injection efficiency depends on the strength of heme interaction between PpcA-E and OmcS nanowires

To determine how the heme interface between PpcA-E and OmcS nanowires affects electron transfer efficiency, we probed the interaction among their hemes directly by comparing the chemical shift of the heme methyl signals of PpcA-E with and without OmcS (1:4 PpcA-E:OmcS heme ratio) (Fig. 4c). The heme methyls are easy to probe as they are in an uncrowded region of the spectrum in the oxidized state. The most affected heme substituents were: PpcA's $2^1CH_3^{III}$, $12^1CH_3^{III}$, $7^1CH_3^{IV}$, $12^1CH_3^{IV}$; PpcB's $12^1CH_3^{I}$ and $18^1CH_3^{I}$; PpcC's heme I substituents and $2^1CH_3^{IV}$ and $18^1CH_3^{IV}$; PpcD's $2^1CH_3^{I}$, $12^1CH_3^{IV}$, and PpcE's heme I substituents and $7^1CH_3^{IV}$, $12^1CH_3^{IV}$. The overall positive surface charge of the PpcA-E[40] and their smaller size in comparison with OmcS (Fig. 7a, b) may favor the interaction between the redox pair and

explain the observation of several affected heme substituents rather than a single one.

The variation in chemical shifts ($\Delta\delta$) indicates whether the interaction between two proteins involves a well-defined interaction surface or is highly dynamic, and there are many conformations that the partner proteins adopt. We observe a slight variation in the chemical shifts (<0.15 ppm) compared to other cytochrome complexes[41]. Therefore, Ppc-OmcS interaction is transient. This transient nature is essential in electron transfer reactions. Notably, $\Delta\delta_{PpcC}$ (0.08–0.14 ppm) was significantly higher than $\Delta\delta_{PpcABDE}$ (<0.04 ppm) (Fig. 4c), indicating a slightly more specific interaction of PpcC with OmcS, likely due to larger binding interface. This interaction explains the highest electron transfer of PpcC among paralogs despite its lowest periplasmic abundance[36].

### Electron injection occurs via transient binding to OmcS nanowires via complementary-charged residues

To further understand how PpcA-E inject electrons into OmcS nanowires, we analysed the surface charges of PpcA and OmcS (Fig. 7). OmcS has several positive and negative surface charges while PpcA is

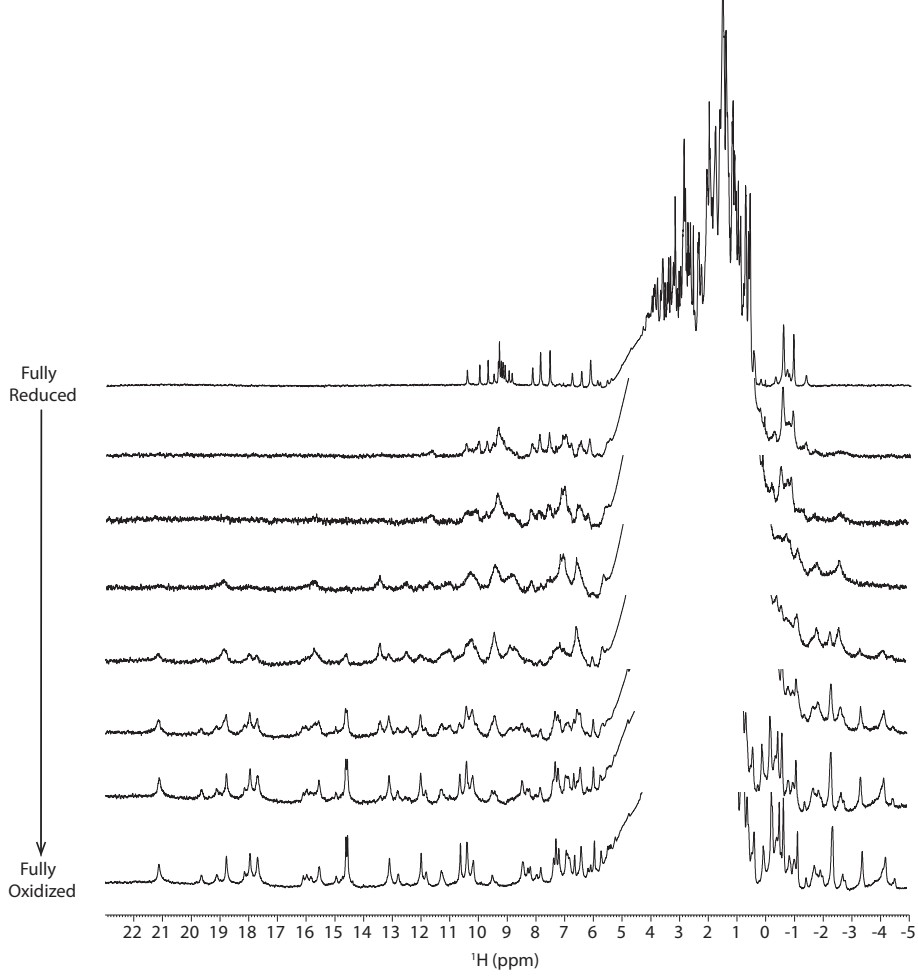

**Fig. 6 | Reference NMR spectra of air-oxidized PpcA upon redox titration.** The spectra of PpcA (60 μM) are pictured in order from the fully reduced state (top) to the fully oxidized state (bottom). The PpcA sample was prepared in 50 mM potassium phosphate buffer with KCl (final ionic strength of 150 mM), pH 7, 100% $^2H_2O$. Spectra were acquired at 298 K, 600 MHz.

primarily positively charged (Fig. 7a, b). Therefore, the most likely region of binding interaction is near an exposed heme on OmcS surrounded by a negatively charged area (Fig. 7a). Therefore, PpcA-E could inject electrons into OmcS by using these complementary surface charges combined with surface-exposed hemes (Fig. 7a, b).

To directly evaluate this possibility, we focused on analysing the change in PpcA backbone N-H signals upon binding to OmcS nanowires (1:4 PpcA:OmcS heme ratio) (Fig. 7c). We chose PpcA because it is important for EET and is the most abundant cytochrome in the periplasm capable of interacting with other carriers[36]. The most affected residues were determined using amino acid-specific chemical shift mapping[42] (see Methods for details). Analysis of the PpcA backbone residues affected by the presence of OmcS ($\Delta\delta_{combined} > \Delta\delta_{cut-off}$) shows two different extents of variation: residues Ile[4], Ala[8], Val[13], Lys[14], Phe[15], Lys[18], Ala[46], His[47], Gly[48] are the most affected ones by the presence of OmcS ($\Delta\delta_{combined} > 1.5\Delta\delta_{cut-off}$), followed by Asp[3], His[20], Cys[30], His[31], Lys[37], Lys[52], Cys[54], Thr[63] and Cys[68] ($\Delta\delta_{combined} > \Delta\delta_{cut-off}$). The most affected residues comprise a region near heme IV (Ala[8], Val[13], Ala[46], His[47], Gly[48]) and in between hemes I and III (Ile[4], Lys[14], Phe[15], Lys[18]) (Fig. 7d). Thus, our studies indicate that PpcA binds to OmcS in these regions via complementary surface charges. Notably, we find that the interaction does not require any specific interaction interface. Thus, periplasmic cytochromes can use an ensemble of protein orientations to achieve efficient electron injection into OmcS.

## Minimal electron transfer pathway for charging OmcS nanowires via PpcA-E

*G. sulfurreducens*' biofilm redox response is centered at −150 mV, highlighting the central role of the periplasmic cytochrome family in the electron transfer chain[20]. However, at the time of the study, the reduction potential value of OmcS (−212 mV) suggested that the electron transfer to OmcS was not thermodynamically favorable[24]. Our spectroelectrochemical analysis on intact OmcS nanowires in this work showed an 82 mV more positive reduction potential with a 40 mV larger redox-active window than previously reported. The larger overlap in potentials indicates OmcS could receive electrons from PpcA-E, suggesting a previously unknown EET pathway with minimal components that could provide a more direct pathway for extracellular electron transfer (Fig. 8). NMR data confirmed electron transfer between PpcA-E and OmcS (Figs. 4 and 5) and showed that PpcA-E interacts with OmcS by binding transiently via complementary-charged residues. The location of the charged residues and docking simulations suggest that transient binding achieves proximity of hemes (Figs. 7d, 8b inset). This direct binding of PpcA-E to OmcS eliminates any need for relying on slow electron diffusion among periplasmic carriers[19,20,23]. It explains why having the OmcS nanowire and only one of the PpcA-E paralogs is sufficient to achieve EET[36,37]. OmcS remains in the periplasm when not fully assembled[10].

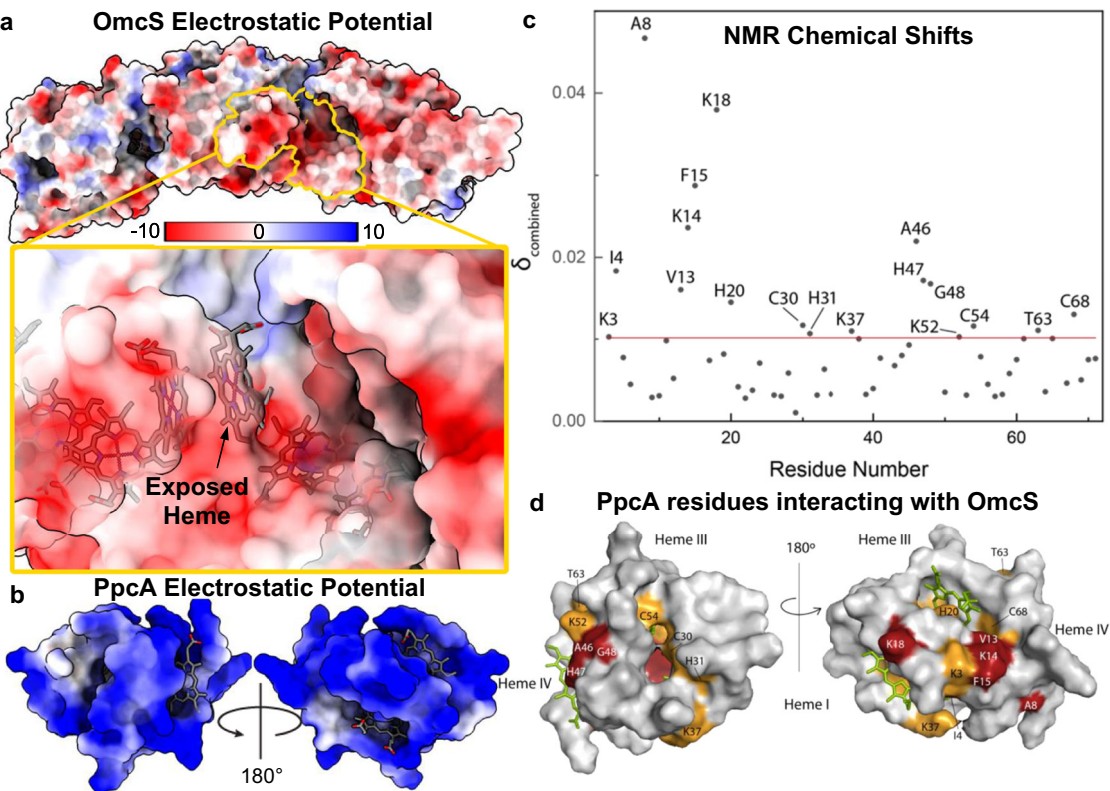

**Fig. 7 | PpcA binds transiently to OmcS nanowires via complementary charge interaction.** Electrostatic potentials of **a** OmcS nanowires with a zoomed region in yellow and **b** PpcA. Both models are shown with color ranging as in the scale bar in (**a**) with units of kT/e at T = 298 K. **c** Combined chemical shift differences of PpcA NH groups (black circles). The cut-off line is in red. **d** Most affected PpcA backbone residues in the presence of OmcS. The residues with $\Delta\delta_{combined} > 1.5\Delta\delta_{cut\text{-}off}$ are colored in red, and the remaining are colored in orange in PpcA (PDB ID: 2MZ9). The hemes are colored in green.

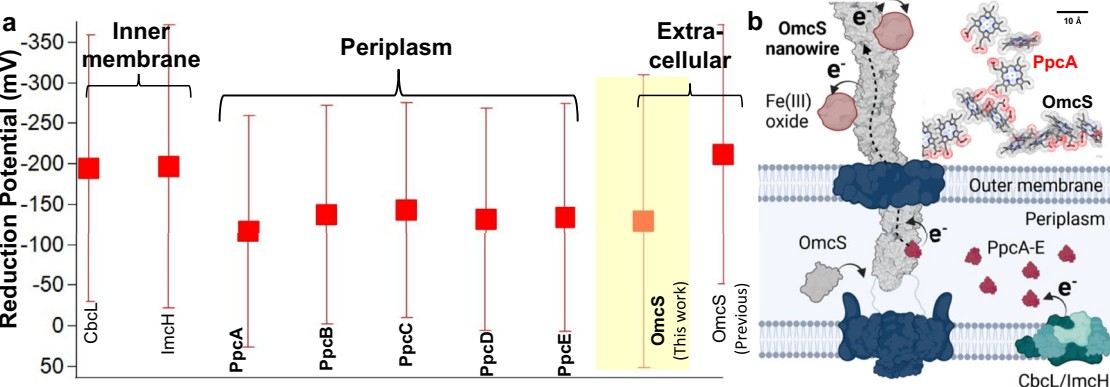

**Fig. 8 | Nanowire-charging pathway. a** Comparison of the redox-active windows of *G. sulfurreducens* cytochromes determined from 1–99% reduction. The cellular localization of cytochromes is indicated above the graph. Red squares represent $E_{app}$ and bars represent redox-active window. **b** Model of EET where PpcA-E receives electrons from the inner-membrane complex and donates them to OmcS nanowires, transporting electrons to extracellular acceptors such as Fe(III) oxide. Created with BioRender.com. Inset, the docking model of PpcA binding to OmcS nanowire suggests the proximity of surface-exposed hemes for efficient electron injection from PpcA to OmcS. Scale bar, 10 Å.

Therefore, our model assumes that nanowire formation initiates in the periplasm itself (Fig. 8b).

**OmcS nanowire-charging pathway via PpcA-E is widespread in phylogenetically diverse EET- and DIET-performing microbes**
We found that diverse environmentally important bacteria possess genes homologous to those encoding PpcA-E and OmcS nanowires, suggesting that the nanowire-charging mechanism is widespread and conserved across microbial kingdoms (Fig. 9). For example, in addition to diverse *Geobacter* species, homologs of both OmcS and PpcA-E were present in thermophiles that live in diverse environments such as *Geoalkalibacter subterraneus* isolated from an oilfield at 1.5 km depth[43] that are capable of EET to iron oxide and manganese oxide[43], and for EET to electrodes to generate high current densities (4.7 A/m²)[44], which requires EET over multiple cell lengths by forming thick, conductive biofilms[3,34]. The homologs of both OmcS and PpcA-E were also present in *Anaeromyxobacter dehalogenans* species capable of EET for use in bioremediation of toxic materials[45–47] (Fig. 9). Natural methanogenic

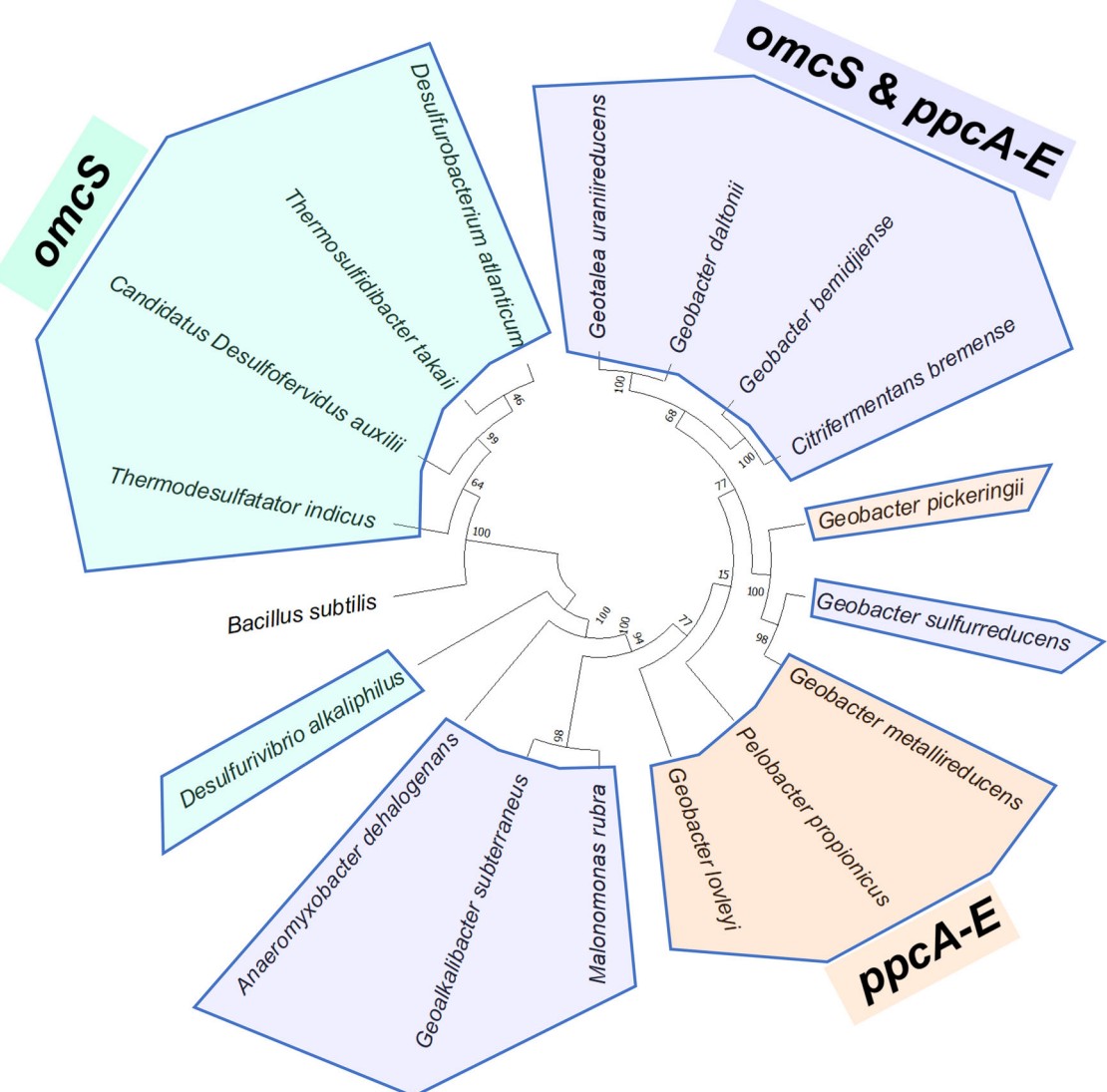

**Fig. 9 | The OmcS nanowire-charging pathway via PpcA-E is widespread in phylogenetically diverse bacteria.** Phylogenetic tree showing PpcA-E and OmcS homologs. Diverse bacteria producing OmcS nanowire homologs also produce PpcA-E homologs (light blue), except for a few marine bacteria (light green) and for a few *Geobacteraceae* (light orange).

archaeal communities collected from diverse geographic locations and in anaerobic methane[48,49] and hydrocarbon[50]-consuming bacterial and archaeal communities[51] are hypothesized to use DIET via nanowires for their growth. Importantly, *Geobacter* has been shown to transfer electrons to methanogens to reduce carbon dioxide to methane[52]. Therefore, our studies suggest that periplasmic cytochromes PpcA-E and OmcS work together to export electrons outside the cells for both EET and DIET, and this previously unknown nanowire-charging pathway is evolutionarily conserved in phylogenetically diverse microbes that regulate our environment, ranging from mesophiles important for global biogeochemical cycling to thermophiles important for bioremediation. Species that lack OmcS show OmcZ that also forms nanowires[8,12], and species that lack PpcA-E could use other carriers such as ExtA[21] and PgcA[36] as alternate EET pathways. Therefore, our approach is widely applicable to studying complex EET pathways, making it possible to assess the diversity of microbes capable of establishing electrical connections via nanowires.

In summary, our studies help to resolve a long-standing mystery of how various soil and marine microbes proliferate in diverse environments by performing EET at remarkably fast rates of million electrons per second[17], which cannot be explained by slow electron diffusion among periplasmic cytochromes[19,20]. We show that PpcA-E can directly transfer electrons and bind to OmcS nanowires. Combining spectroscopy with electrochemistry, we demonstrate that physiological OmcS nanowires have a midpoint potential of −130 mV, which is 82 mV-more positive than in prior reports. The physiologically-relevant reduction potential and wide redox-active window of OmcS is ideally suited for accepting electrons from diverse intracellular electron carriers (Fig. 8a).

To move electrons from the inner membrane to the outer surface, periplasmic cytochromes bind transiently to nanowires via complementary-charged residues without the need for a diffusive network of cytochromes as long thought[23]. As we find that all PpcA-E can inject electrons into OmcS, our studies explain prior genetic studies that only one of the paralogs among PpcA-E is sufficient to restore EET and periplasmic cytochromes can serve as capacitors to store electrons when EET is not feasible[53]. Furthermore, we found that PpcC, which has the most negative midpoint reduction potential, shows the highest electron transfer efficiency to OmcS, which could explain the ability of PpcC to restore EET to wild-type levels despite its low protein abundance[36].

Prior genetic studies have shown that deleting the most abundant genes encoding outer-surface and periplasmic cytochromes, except PpcA and an OmcB porin-cytochrome complex, still enables EET to Fe(III) citrate. However, this is a soluble iron, which is reduced on the bacterial surface and thus only limited to short range[37]. Our studies suggest that OmcS nanowires and any of PpcA-E paralogs are sufficient to reconstitute minimal machinery capable of long-range EET to insoluble Fe(III) oxides as well as DIET as both these processes require OmcS nanowires. Identifying the essential components necessary for long-range electron export will help develop additional microbial chassis capable of EET and DIET for various applications, such as bioelectronics and biofuel production[6].

Complementary surface charges and heme-to-heme proximity guide the interaction between PpcA-E and OmcS. Binding to the negatively charged interface of the nanowire (Fig. 7a, b) could be more advantageous than binding to another region of the monomer because such a binding pocket would not exist for monomeric OmcS, thus making the interaction specific to the nanowire and limit non-specific interaction between PpcA-E and OmcS monomers. Building on our studies to couple electron and proton transfer in PpcA-E via their rational design[14,54], future studies to rationally engineer the interface between nanowires and PpcA-E could enable the control of electron injection rate by bacteria during EET and DIET. Such design approaches could enable control over EET and DIET rates to improve microbial performance.

Remarkably, this defined nanowire-charging pathway is widespread in diverse environmentally important bacteria, including thermophile bacteria and bacteria critical from bioremediation of oil spills and toxic materials (Fig. 9). Our studies show that the periplasmic cytochromes and nanowires co-occur in such bacteria growing in extreme environments, enabling bacteria to export electrons rapidly, without relying on the slow diffusion of monomeric or soluble electron carriers. This efficient EET pathway could explain enhanced electricity production in genetically engineered strains that overproduced OmcS nanowires[11,55] via forming highly conductive biofilms[3,55–57] and show large supercapacitance to store electrons when electron acceptors are not available[53]. Notably, OmcS homologs are also highly expressed in DIET-performing co-cultures oxidizing methane[3,48,58], and cytochromes abundant in the subsurface during uranium bioremediation function similar to OmcS[59]. Methane-oxidizing archaea primarily show OmcZ nanowire homologs whereas diverse bacteria show OmcS nanowire homologs[8,60]. Our identification of a widespread EET pathway thus provides direct insight into how diverse microbes proliferate at rates too fast for diffusion by rapidly transferring electrons. Furthermore, our NMR-based approach to quantify electron injection is widely applicable to studying complex pathways, making it possible to assess the diversity of microorganisms capable of establishing electrical connections via various proteins.

Given that methane is a significant contributor to climate change and that nanowire-producing microbes impact a significant portion of methane levels[48,49,51,52], future studies focused on understanding how these microbes use PpcA-E and OmcS homologs together could result in additional methods for controlling DIET in microbial communities to reduce global methane emissions, and thereby to combat climate change.

## Methods

### Bacterial growth conditions and filament preparation
Cells from bacterial strain $\Delta omcZ$[61] were grown in 10 L jugs at 25 °C to $OD_{600}$ ~ 0.8–1 in NBAF media as described previously[62] with ten-times more $MgSO_4$. Cells were collected at ~15,000 × $g$ for 12 min. Pellets were resuspended in 150 mM ethanolamine pH 10.5, and homogenized with a tissue grinder then left to stir at 4 °C overnight. Cells were then blended on low speed for 2 min, and cell debris was pelleted by spinning at 10,000 × $g$ for 30 min. The supernatant was homogenized by stirring overnight with 1 ml 12.5% Triton X-100 per 250 ml supernatant. The solution was then dialyzed to 20 mM triethanolamine pH 8 exhaustively using 50 kDa cutoff membrane. Nanowires were pelleted by spinning solution at 23,000 × $g$ for 1 h. Pellets were resuspended in 20 mM ethanolamine pH 10.5 and left on a rotary shaker overnight. Any insoluble material left was removed by centrifugation at 10,000 × $g$ for 30 min. This process was repeated as needed. The sample was run through a gel filtration column Sephacryl S–500 HR, and fractions were pooled to obtain pure nanowires. Fractions were concentrated as needed using the 20 mM triethanolamine pH 8 precipitation protocol outlined above.

### Negative staining transmission electron microscopy (TEM)
Electron microscopy science grids (mesh size 400) were plasma cleaned for 30 s in PDC-001-HP Harrick Plasma cleaner. About 5 μl of sample (nanowires or cell culture) was dropped onto the carbon face of the grid and left to stand for 10 min. Excess solution was blotted. Samples were placed face down on 50 μL drops of negative stain (1% phosphotungstic acid, pH 6) for 30 s; the remaining solution was removed by blotting, then repeated. Grids were air-dried and stored in a sealed case until imaging using a JEM-1400Plus microscope operating at 80 kV (JEOL).

### Spectroelectrochemistry
Spectroelectrochemistry experiments were performed in a Honeycomb Spectroelectrochemical Cell (Pine Research Instrumentation, Durham, NC)[30]. Briefly, 1 ml of purified OmcS was concentrated through a 30 kDa cutoff Amicon centrifugation column at 14,000 × $g$ for 5 mins. 400 μl electrolyte solution (50 mM $KH_2PO_4$ and 100 mM KCl at pH 7) was added to the column, and any pelleted nanowires were resuspended. This process was repeated 3 times. A final spin at 17,000 × $g$ was used to pellet nanowires into an Eppendorf. Any excess buffer was pipetted off, leaving a pellet of nanowires. The tube was put into an anaerobic chamber with the cap open, and ambient oxygen was removed by the transfer purge process. The pellet was resuspended in 200–400 μl of nitrogen-purged electrolyte until homogenous. The concentration of OmcS was measured by absorbance at the 528 nm peak (α-peak) and was between 30–60 μM for all trials[22]. A solution of equal concentrations of 15 mediators that span a potential range from −440 to +80 mV $vs$ SHE was added to a final concentration of 1–2% of the heme concentration in the sample[22]. The concentration of mediators was optimized by selecting the highest [Mediator]/[Hemes] percentage that did not distort the α-peak despite background subtraction (Supplementary Fig. 2). Subsequent potential steps were applied to the sample for 2 min, and simultaneous UV-visible spectra were recorded. Data were scanned in the cathodic and anodic direction while mixing the nanowire samples between runs.

There was some hysteresis observed, with the anodic trace being, on average, more positive than the cathodic by 17 mV. The hysteresis might indicate some changes in protein structure accompanied by the redox transition[63], or it could be from the large macroscopic nature of the nanowires or aggregation, which might influence reaction kinetics at the electrode surface.

### Data analysis
Spectra were recorded continuously throughout the measurements. The time signatures of the applied potentials and spectra were synchronized to process only the spectra recorded at the end of the 2-minute window. Isosbestic points were calculated by finding the points with the lowest standard deviation across the collected spectra in the expected range (540–565 nm). The area under the α-peak above the line between isosbestic points was used to calculate the oxidized fraction. Midpoint potential was determined as described previously[64] by fitting a single Nernst equation to the data to compute the oxidized fraction (Eq. 1) where $v1$–$v6$ are reduction potentials of six hemes

computed using SEC (Fig. 2d). Individual potentials were fit with independent or sequential six-centered Nernst equations to compute oxidized fractions (Eqs. 2–5), where $F$ is the Faraday constant, $R$ is the gas constant, and $T$ is the temperature.

$$P_{ox}(x) = \frac{e^{(x-v1)k}}{\left(1 + e^{(x-v1)k}\right)} \tag{1}$$

$$P_{ox}(x) = \frac{1}{6}\sum_{i=1}^{6} \frac{e^{(x-v_i)\frac{F}{RT}}}{\left(1 + e^{(x-v_i)\frac{F}{RT}}\right)} \tag{2}$$

$$p_r = 6 + 5e^{(x-v_1)\frac{F}{RT}} + 4e^{(2x-v_1-v_2)\frac{F}{RT}} + 3e^{(3x-v_1-v_2-v_3)\frac{F}{RT}} \cdots \\ + 2e^{(4x-v_1-v_2-v_3-v_4)\frac{F}{RT}} + e^{(5x-v_1-v_2-v_3-v_4-v_5)\frac{F}{RT}} \tag{3}$$

$$Z = 6\big(1 + e^{(x-v_i)k} + e^{(2x-v_1-v_2)\frac{F}{RT}} + e^{(3x-v_1-v_2-v_3)\frac{F}{RT}} \\ + e^{(4x-v_1-v_2-v_3-v_4)\frac{F}{RT}} \cdots + e^{(4x-v_1-v_2-v_3-v_4)\frac{F}{RT}} \\ + e^{(5x-v_1-v_2-v_3-v_4-v_5)\frac{F}{RT}} + e^{(6x-v_1-v_2-v_3-v_4-v_5-v_6)\frac{F}{RT}} \tag{4}$$

$$P_{ox}(x) = 1 - \left(\frac{p_r}{Z}\right) \tag{5}$$

Spectroelectrochemical data can be fit with independent or sequential redox centers, where the sequential model implies interaction between hemes, and the independent model does not. We evaluated fitting the experimental data (Fig. 2c) with six independent and six sequential Nernstian centers, which yielded comparable results. The close agreement between the experimental macro- and microscopic potentials suggested that a single heme contributed to each redox transition between the different oxidation stages in OmcS. This dominant contribution of a single heme suggests that the heme-heme interactions were negligible compared to the uncertainties in the reported potentials.

## Electrochemistry

Devices were fabricated as described previously[35]. PDMS wells with an outer diameter of 9 mm and inner diameter of 6 mm were attached to each device around the interdigitated array by ambient air plasma cleaning both the device and PDMS well on medium for 45 s. Then, the well was placed on the device and pressed firmly together. Platinum wire with a 0.5 mm diameter served as the counter electrode, while an Electrolytica C-925 1 mm (Ag/AgCl with 3.4 M KCl) served as the reference electrode. The reference was compared to a standard before each use. The counter electrode was cut fresh or cleaned in 0.5 M $H_2SO_4$, as were the gold electrodes before use by holding 2.04 V for 5 s, −0.31 V for 10 s, followed by cyclic voltammetry from −0.26 V to 1.6 V at 4000 mV/s for 20 cycles then at 100 mV/s for 4 cycles[65]. Electrodes were washed in ethanol, then water, and dried in a stream of $N_2$. 1.5 μL of filament solution was drop cast onto the device and air-dried overnight before measurement. Cyclic Voltammetry (CV) and Differential Pulse Voltammetry (DPV) were performed using Gamry potentiostats. CV was performed for all films from 0.4 to 0.7 with a step size of 1 mV at a scan rate of 10 mV/s unless otherwise indicated. All DPV were performed from 0.4 to −0.7 with a step size of 1 mV, sample period of 0.5 s, pulse size of 25 mV and pulse time of 0.1 s. The electrolyte in all cases was 50 mM $KH_2PO_4$ and 100 mM KCl at pH 7, sparged with $N_2$ for at least 30 min and stored in a serum tube to remain anaerobic as described previously[62]. Peak fitting was performed using MATLAB scripts, including the EzyFit and peakfit modules.

## NMR sample preparation and spectra acquisition

PpcA-E periplasmic cytochromes were overexpressed and purified according to the protocol described previously[66]. Briefly, competent *Escherichia coli* BL21(DE3) cells containing the pEC86 plasmid—which codifies for the cytochrome maturation system *ccmABCDEFGH* and has resistance to chloramphenicol—were transformed with a plasmid containing either *ppcA*, *ppcB*, *ppcC*, *ppcD* or *ppcE* genes, with resistance to ampicillin, following the heat shock method. The proteins' overexpression was performed in 2xYT media supplemented with ampicillin (100 μg/mL) and chloramphenicol (34 μg/mL). A colony was inoculated in 50 mL of liquid 2xYT media for the starter culture and grown for 18 h at 30 °C, 200 rpm. On the next day, 10% of the starter culture was inoculated in 1 L of 2xYT media and the cells were incubated at 30 °C, 180 rpm, until they reached an $OD_{600nm}$ ~1.5. At this moment, protein production was induced with 10 μM isopropyl-β-D-thiogalactoside (IPTG), the rotation was lowered to 160 rpm, and the cells were incubated for 18 h. Cells were collected by centrifugation at $6400 \times g$, 20 min, 4 °C. The cell pellet was resuspended in 30 mL lysis buffer [20% sucrose (Fisher Scientific), 100 mM Tris–HCl pH 8.0 (NZYTtech), 0.5 mM EDTA pH 8.0 (Sigma), 0.5 mg/mL lysozyme (Fluka)] per liter of initial cell culture. The lysate was centrifuged twice, at 15,000 g, 20 min, 4 °C and then at $150,000 \times g$, 90 min, 4 °C. The red supernatant was dialyzed twice in 10 mM Tris–HCl pH 8.5 buffer and was purified first using ion-exchange chromatography (2×5 mL Bio-Scale™ Mini UNOsphere™ S Cartridges [BioRad]) equilibrated with the same dialysis buffer and eluted with a 150 mL gradient from 0–300 mM NaCl. The red fractions containing the protein were pooled together, concentrated to less than 2.5 mL and loaded onto a Superdex 75 XK16/60 molecular exclusion column (GE Healthcare) equilibrated with 100 mM sodium phosphate pH 8.0. Protein purity was confirmed through SDS-PAGE gel stained with BlueSafe (BioRad), and protein integrity was confirmed by acquiring 1D $^1$H- NMR spectra (acquisition parameters described below).

The proteins were lyophilized thrice and resuspended in a 50 mM potassium phosphate buffer with KCl (final ionic strength of 150 mM), pH 7, and prepared in $^2H_2O$. OmcS (30 μM) was exchanged to Mili-Q water using amicon filtration units with 30 kDa cut-off (Merck) and subsequently lyophilized once. As described previously[39], oxidized periplasmic cytochrome (PpcA-E) sample (60 μM) was reduced inside an NMR tube by flushing out all the sample oxygen with hydrogen and adding catalytic amounts of hydrogenase from *Desulfovibrio vulgaris* (Hildenborough)[67]. OmcS was added anaerobically to each periplasmic cytochrome in heme proportions 1:1 and 1:4 in a glovebox (MBraun). The NMR tube was sealed with a rubber cap, and the 1D $^1$H- NMR spectrum was immediately acquired. For the control experiment, degassed potassium phosphate buffer was added to a reduced PpcA-E instead of OmcS. At the end of the experiment, the NMR tube containing the periplasmic cytochrome and OmcS in a 1:4 heme ratio was opened, and the NMR spectrum was acquired.

For probing the interaction interface between the pairs of proteins, the 1D $^1$H- NMR spectrum of oxidized PpcA-E was acquired with and without OmcS (1:4 heme ratio periplasmic cytochrome:OmcS), and spectra were compared to measure the perturbation on the periplasmic cytochrome's heme chemical shifts. The heme chemical shifts were previously assigned and are reported in Supplementary Table 1.

For interaction experiments in the oxidized state, isotopically labeled $^{15}$N PpcA (overexpressed and purified according to the protocol discussed previously[68]) was resuspended in 8 mM potassium phosphate buffer with KCl (20 mM final ionic strength), pH 7, prepared in $H_2O/^2H_2O$ (92%/ 8%) and NMR spectra were acquired in the absence and presence of OmcS (PpcA:OmcS heme ratio 1:4). PpcA's backbone assignment in the absence and presence of OmcS is reported in Supplementary Table 2.

All NMR spectra were acquired in a Bruker Avance 600 MHz spectrometer at 298 K. Pulse calibration and standard experiments

performed using Bruker's available library. All 1D $^{1}$H- NMR spectra were acquired with 3.3 k data points and a spectral width of 24 kHz, with 256 scans. For backbone interaction experiments, 2D $^{1}$H, $^{15}$N- HSQC spectra were acquired with 2048 ($t_2$) × 256 ($t_1$) data points, with a sweep width of 11.4 k ($^{1}$H) and 2.4 k ($^{15}$N) and 32 scans. $^{1}$H chemical shifts were referenced to 2,2-dimethyl-2-silapentane-5-sulfonate at 0 ppm[69]. The NMR spectra were processed and analysed using TOPSPIN 3.6.5 software (Bruker Biospin).

## Chemical shift analysis

As described previously[42], the weighted average chemical shift of the backbone NH signals of an amino acid $j$ ($\Delta\delta_{combined}$) was determined as: $\Delta\delta_{combined,j} = \sqrt{(\Delta\delta_H)^2 + (w_i\Delta\delta_N)^2}$ where $\Delta\delta_H$ and $\Delta\delta_N$ correspond, respectively, to the chemical shift change in the $^{1}$H and $^{15}$N dimensions in the absence and presence of OmcS and $w_i$ is the quotient between the gyromagnetic ratio of $^{15}$N and $^{1}$H to compensate for scaling differences[42]. The most affected residues were selected using a threshold based on the procedure described previously[42]: the cut-off value was determined iteratively by determining the standard deviation, $\sigma_{corr}$, from the dataset and excluding the residues which presented $\Delta\delta_{combined}$ three times higher than the calculated $\sigma_{corr} = \sqrt{\frac{1}{N}\sum(\Delta\delta_{combined} - 0)^2}$ where N is the number of residues used. This process was iteratively repeated, calculating a new $\sigma_{corr}$ using the remainder $\Delta\delta_{combined}$ values until no more residues were excluded. The cut-off value ($\Delta\delta_{cut-off}$) chosen as a criterion corresponded to the last calculated $\sigma_{corr}$ and has the value of 0.01.

The same procedure was applied to calculate the most affected heme methyl signals. In this case, the $\Delta\delta_{combined}$ corresponds to the difference of the heme methyl chemical shifts in the presence and absence of OmcS. The determined cut-off values were 0.005 ppm (PpcA), 0.010 ppm (PpcB), 0.084 ppm (PpcC), 0.007 ppm (PpcD), 0.009 ppm (PpcE).

## Visualization of electrostatic potential and docking

Electrostatic potential and docking for OmcS (PDB ID: 6EF8) and PpcA (PDB ID: 2MZ9) were generated using ChimeraX and ZDOCK software, respectively.

## Bioinformatic analysis

The phylogeny tree of the housekeeping gene *gyrB* was constructed using MEGA X software's maximum likelihood method[70]. OmcS and PpcA-E homologs are obtained from NCBI Reference Sequence Database (Refseq) with expectation (e)-value $< 1e^{-5}$ and coverage higher than 75%.

## Reporting summary

Further information on research design is available in the Nature Portfolio Reporting Summary linked to this article.

## Data availability

All data sets presented in the study are included in the paper. Source data are provided in this paper.

## Code availability

MATLAB code used to analyse spectroelectrochemistry data is available at github [https://github.com/ccshipps/OmcS_Redox].

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

## Acknowledgements

We thank Gary Brudvig for advice on spectroelectrochemistry measurements and M. Gubermann-Pfeffer for help with ZDOCK. This research was supported by the National Defense Science and Engineering Graduate Fellowship (to C.C.S.), the Human Frontier Science Program award no. RGP017/2023 (to N.S.M. and C.A. S.), NSF CAREER award no. 1749662 (to N.S.M.), the NSF-ANR award no. 2210473 (to N.S.M.), and the National Institutes of Health (NIH) Director's New Innovator award (1DP2AI138259-01 to N.S.M.). NMR work was supported by Fundação para a Ciência e Tecnologia (FCT) through PTDC/BIA- BQM/4967/2020 (to C.A.S), 2020.04717.BD (to P.C.P) and a 2022 Fulbright Scholarship with the support of FCT (to P.C.P). The Applied Molecular Biosciences Unit – UCIBIO, which is financed by national funds from FCT (UIDB/04378/2020 and UIDP/04378/2020), and the project LA/P/0140/2020 of the Associate Laboratory Institute for Health and Bioeconomy – i4HB also supported this work. The NMR spectrometers are part of the National NMR Network (PT NMR) and are supported by FCT- MCTES (ROTEIRO/0031/2013 – PINFRA/22161/2016) co-funded by FEDER through COMPETE 2020, POCI, and PORL and FCT through PIDDAC.

## Author contributions

P.C.P. purified PpcA-E and OmcS nanowires and performed NMR. C.C.S. purified OmcS nanowires and performed biochemical analyses, spectroelectrochemistry and electrochemistry. C.S. performed bioinformatic analysis. V.S. performed electron microscopy and initial modeling. C.A.S. supervised NMR work. N.S.M. supervised the project and wrote the manuscript with input from all authors.

## Competing interests

The authors declare no competing interests.
