## [Peer Review File · Nature Communications]

Reviewer #1 (Remarks to the Author):

Points on nature comms paper

Overview remarks

This is a nice and interesting paper that uses a variety of techniques, including a nice methodological approach with NMR to look at interaction and electron transfer between Ppc cytochromes and OmcS in Geobacter. There are some novel observations.

I would say though that I think some citations of earlier work are slightly misrepresented. For example, it is stated that there is a 'near universal view' that periplasmic cytochromes exchange electrons with each other. Only one paper is cited to justify this sweeping statement - a review / commentary paper that is almost 20 years old and really does not focus on cytochromes and intermembrane electron transfer very much at all. I have been in the field 30 years and I don't think this is a universal view

Sections

Purification of OmcS nanowires...

This section seems fine. The SDS PAGE seems quite overloaded and still only shows one band for example and the TEM looks okay.

Spectroelectrochemistry...

The revised mid-point potential seems fine and important to have done this as it does seem that previous work was not on intact OmcS nanowires and the discussion of this seems fair. However, I don't think that the current published value has left people thinking that the Ppc cytochromes do not pass electrons to OmcS - the thermodynamics were more challenging with the 'old value', but not prohibitory so long as there is strong driving force and a strong 'pull'. For example Ref 31 paper I don't think specifically excluded this possibility.

OmcS nanowires show nearly thermoneutral energy landscape...

I am not sure I quite follow the data or arguments here. It is stated that all six heme in OmcS showed reduction potential with tens of mV from each other. But this is not the case. Redox component V6 Em is for example is some 200 mV lower than V1. This does not feel like a thermoneutral landscape to me.

Also I don't think you can know that in the structure, for example V2 is adjacent to V1 or V3 so it is not possible to be sure that adjacent hemes have Ems within 10 mV of each other.

Also although they suggest this situation is different to the MtrC family (e.g. MtrC, OmcA, MtrF). I don't think it is. For example the situation for MtrF (from PNAS - Clarke et al. www.pnas.org/cgi/doi/10.1073/pnas.1017200108) seems similar

I may be missing the argument though. Finally, it is stated the MtrC is monomeric. It isn't - it forms a hardwired 20 heme complex with MtrAB.

Surface-adsorbed OmcS...

This section seems fine and it is good to compare solution state versus solid state.

NMR shows....and Electron efficiency sections...

These are a strong sections and nice approaches to studying protein-protein interactions and electron transport. The assignment of the NMR signals is good.

Electron injection occurs via...

For me it is nice to try to find possible molecular specificity and use terms like 'electron injection', and I think this section is fine. But personally, I don't think electrons really are 'injected'. It is not that precise. It is electron transfer. Electrons will always find a way to flow if the driving force is there, so I prefer to use the term electron transfer.

OmcS nanowire charging via...

This is a nice bioinformatics section and is interesting and suggestive of the more widespread importance of this work.

Finally, – the periplasm is not a space, but this term is used a few times. It is a subcellular compartment. In one section the authors say it is a 'space' and then say it is a 'crowded' environment (lines 44-47)! A 'space' can't be crowded so I prefer to call it the periplasmic compartment. Actually, some 35 years ago it was shown it likely has a texture of thick porridge –it is indeed a crowded environment.

Reviewer #2 (Remarks to the Author):

Report on manuscript no. NCOMMS-23-36005, entitled “A widespread extracellular electron transfer pathway for charging microbial 1 cytochrome OmcS nanowires via periplasmic cytochromes PpcABCDE” by P.C. Portela et al.

The authors have taken on a challenging chemical problem in a biological system that has widespread interest for those studying electron transfer in Nature. The manuscript is relatively densely written, although there is a lot of ground to cover to ensure the reliability of the interpretations, which do appear plausible and correct. Some statements are unclear. However, some clarifications and reordering of material might make the ultimate paper more accessible to the reader. Some suggestions and some other questions arising from the manuscript are listed below. If the authors respond satisfactorily to these then I will be pleased to recommend this work for publication in Nature Communications.

- 1) P 1, l 37, refer to Figure 1(b) at the first mention of the nanowires to make this report more accessible to readers, and place this figure closer to its first call from the text.
- 2) P 2, l 52, reference to another multi-haem species that has had a reduction potential measured in both (near) native and isolated environments would support this highly likely assertion.
- 3) P 2, l 82, reference 8 does not explicitly mention the Cotton effect, but appears to use changes in CD spectra and comparisons with calculations using density functional theory to draw conclusions about their nanowires. Some further clarification on this point might aid the reader.
- 4) P 2, Figure 1(b), here and elsewhere there is no mention of inter-haem geometry and any requirements of that to facilitate electron transfer.
- 5) P 3, l 85, the statement about potentials lying within tens of millivolts is not supported by the content of Figure 2(d), and there appears to be a 1-2-2-1 pattern of grouping of potentials.
- 6) P 3, l 93, it is unclear if there was any microscopy performed (if it were possible) on the nanowires that were adsorbed on gold to ascertain location and orientation.
- 7) P 3, ll 110-11, 50 kD and 10 kDa and the 1,000-fold multiplier are unclear.
- 8) P 4, Figure 3, labelling is difficult to read.
- 9) It is unclear how to compare the voltages in Figure 2(d), in Supplementary Table 1 and in Figure 5.
- 10) P 4, l 117, “As OmcS monomer contains six hemes and each PpcA-E contain 3 hemes,” – the compositions do not seem to have been set out clearly early on in the manuscript and are confusing at this point in the development of the science. This also affects understanding the argument on line 123 about the heme ratio.
- 11) P 4, Figure 3, caption, the oxidized form should be light blue and not blue?

- 12) P 5, l 146, “the chemical shifts were within the reported for labile complexes, an essential characteristic in electron transfer reactions.” – meaning is unclear.
- 13) P 5, caption to Figure 4, this is confusing: “with units of kcal/e• mol” – what is this? “Combined chemical shift” – what is this? “highest $\Delta\delta_{\text{combined}}$ ” how is this defined?
- 14) P 6, l 161, “the most affected residues” – how has this been quantified, what criteria were used?
- 15) P 6, l 166, “An 82 mV- positive”, what does this mean?
- 16) P 6, caption to Figure 5, “determined from 1-99% reduction/oxidation.” – what does this mean?
- 17) P 6, caption to Figure 5, “suggests the stacking of surface-exposed hemes for efficient electron injection” – what does this mean?
- 18) P 7, ll 209-10, “This contrasts with the long-standing model that EET requires periplasmic cytochromes exchanging electrons among each other.” In the new model presented here, is this exchange still possible, and from what has now been revealed about binding is there a consequence for cytochromes residing in the periplasm?
- 19) P 8, there seems to be some repetition in the text around here, the very neat new results are in danger of being obscured. Note also that a “Third” point is made twice (lines 247 and 248).
- 20) P 8, l 271, “PpcA and a porin-cytochrome complex can be sufficient for EET to soluble iron, which is reduced on the bacterial surface and thus only limited to short range.” – what does this mean?
- 21) P 11, caption, “larges” ?
- 22) Pages 12-13, tidy up the presentation of some of the experimental work: quantities_space_units
- 23) P14, l 424, page numbers 1 – 100 ?

Reviewer #3 (Remarks to the Author):

This is a great manuscript with important information about the electron transfer mechanisms in *Geobacter sulfurreducens*. Learning that periplasmic cytochromes are able to transfer electrons to a cytochrome nanowire opens up possibilities that simplify the current view of how EET occurs. It also opens up a series of questions that, while not the main purpose of the manuscript, would be great for the authors to discuss. I only have a few comments and suggestions on discussion topics that would help improve the manuscript.

- My first thought when reading the title was: “what about the Ext cytochromes?”. These are not only thought to be crucial components of the outer membrane, but in this paper from the Bond lab (10.1128/JB.00347-18) were shown to be needed for metal reduction or anode electron delivery. If nanowires can directly accept electrons from the periplasm, why are these outer membrane cytochromes so crucial in EET? I do not expect the authors to answer this question, but the lack of

discussion of these cytochromes and the current view on how electron transfer takes place (vs. a view where nanowires take electrons from PpcABCDE) is largely missing.

- There is an embedded assumption that OmcS nanowires start at the periplasm in order to interact with PpcABCDE, then crossing the OM. I know this is a theory previously proposed by the Malvankar lab, starting with the studies on PilA. Nonetheless, the importance of this is not mentioned here. As far as I know this is still a theory (no specific evidence of nanowires in the cytochrome periplasmic space), but an important one to discuss in the context of the findings of the manuscript.

- Many publications have shown reduction potentials of *Geobacter sulfurreducens* biofilms, most of which have a midpoint around -150 mV. The low potentials previously reported for OmcS was conflicting with the knowledge of biofilm electrochemistry. In this sense, the new potential reported for OmcS clears up a lot of discrepancies in the field (e.g., expression studies, deletion studies suggesting omcS is associated to a -150mV midpoint potential pathway). I suggest the authors point this out as an important contribution of their work.

- Page 1 ln 47 – I would not dismiss diffusion as “slow” ($10^5/s$) based on calculations of electron transfer between cytochromes (Reference 19). Diffusion is concentration dependent, so a 10x increase in electron rates can easily be achieved by having 10x periplasmic cytochromes to nanowires. In any case, the discussion on redox potentials after this is much more compelling to set the stage for your manuscript.

- Page 3 ln 99 – While I understood the term, I found it inadequate to call the electron transfer by PpcC “efficient” because it was able to relatively transfer more electrons than other Ppc cytochromes. Especially when considering that the higher electron transfer is probably associated to a higher difference in redox potentials, the term efficiency seems weird in this context.

Point-by-point response to all reviewer comments. Comments are in bold.

Reviewer #1 (Remarks to the Author)

Points on nature comms paper

Overview remarks

This is a nice and interesting paper that uses a variety of techniques, including a nice methodological approach with NMR to look at interaction and electron transfer between Ppc cytochromes and OmcS in Geobacter. There are some novel observations.

We are very grateful for these appreciative comments by the reviewer.

I would say though that I think some citations of earlier work are slightly misrepresented. For example, it is stated that it is there is a ‘near universal view’ that periplasmic cytochromes exchange electrons with each other. Only one paper is cited to justify this sweeping statement -a review / commentary paper that is almost 20 years old and really does not focus on cytochromes and intermembrane electron transfer very much at all. I have been in the field 30 years and I don't think this is a universal view

As suggested by the reviewer, we have removed the phrase “nearly universal” and replaced it with widely accepted. We have now referenced three papers 15 years apart that show how a network of periplasmic cytochromes has been a commonly accepted model for electron transfer (References 23, 27, 28). In addition to citing recent review, we have also cited an experimental paper that shows such network in a related organism that lacks OmcS-like nanowires. We hope that this rewritten text clarifies further that all prior models of EET are based on electron hopping between individual monomeric cytochromes in the periplasmic compartment.

Sections

Purification of OmcS nanowires...

This section seems fine. The SDS PAGE seems quite overloaded and still only shows one band for example and the TEM looks okay.

We thank the reviewer for these appreciative comments.

Spectroelectrochemistry...

The revised mid-point potential seems fine and important to have done this as it does seem that previous work was not on intact OmcS nanowires and the discussion of this seems fair. However, I don't think that the current published value has left people thinking that the Ppc cytochromes do no pass electrons to OmcS -the thermodynamics were more challenging with the ‘old value’, but not prohibitory so long as there is strong driving force and a strong ‘pull’. For example Ref 36 paper I don't think specifically excluded this possibility. As suggested, we have rewritten this section by adding more discussion and citations as follows: Initial models of EET proposed that periplasmic cytochromes do not transfer electrons directly to OmcS and rather require several intermediate cytochromes such as OmcB and OmcE (Nature Rev. Micro. 2006). Prior finding of substantially negative reduction potential of OmcS was noted to be opposite of how electron transfer chains are commonly arranged (Ref. 24: ChemPhysChem, 2011). Ref. 36 does not discuss the possibility of Ppc cytochromes transferring electrons to OmcS. Therefore, we have not included this reference in this section.

OmcS nanowires show nearly thermoneutral energy landscape...

I am not sure I quite follow the data or arguments here. It is states that all six heme in OmcS showed reduction potential with tens of mV from each other. But this is not the case. Redox component V6 Em is for example is some 200 mV lower than V1. This does not feel like a thermoneutral landscape to me. Also I don't think you can know that in the structure, for example V2 is adjacent to V1 or V3 so it no possible to be sure that adjacent hemes have Ems within 10 mV of each other.

As suggested by the reviewer, we have removed the text regarding thermoneutral to avoid confusion.

Also although they suggest this situation is different to the MtrC family (e.g. MtrC, OmcA, MtrF). I don't think it is. For example the situation for MtrF (from PNAS - Clarke et al. www.pnas.org/cgi/doi/10.1073/pnas.1017200108) seems similar

We have clarified this point that the potentials we are referring to are individual potentials of hemes and not average potentials computed in the reference mentioned by the reviewer. Individual heme potentials of MtrC indeed computed to be more than 300 mV different (Fig. 2 in Ref. 32 - Barrozo, A., El - Naggari, M. Y. & Krylov, A. I. *Angewandte Chemie International Edition* **57**, 6805-6809 (2018))

I may be missing the argument though. Finally, it is stated the MtrC is monomeric. It isn't - it forms a hardwired 20 heme complex with MtrAB.

We have clarified that although MtrC itself is monomeric and does not polymerize like OmcS, it does form a porin-cytochrome complex with MtrAB.

Surface-adsorbed OmcS...

This section seems fine and it is good to compare solution state versus solid state.

We thank the reviewer for these appreciative comments.

NMR shows....and Electron efficiency sections...

These are a strong sections and nice approaches to studying protein-protein interactions and electron transport. The assignment of the NMR signals is good.

We thank the reviewer for these appreciative comments.

Electron injection occurs via...

For me it is nice to try to find possible molecular specificity and use terms like 'electron injection', and I think this section is fine. But personally, I don't think electrons really are 'injected'. It is not that precise. It is electron transfer. Electrons will always find a way to flow if the driving force is there, so I prefer to use the term electron transfer.

We have clarified this point by distinguishing between electron transfer and electron transport driven by injection. Electron injection is referring to the specific redox partnership established by PpcA-E and OmcS nanowires that allows electrons to directly flow from periplasmic cytochromes to the nanowire, providing the latter with reducing power that can be then used for extracellular electron acceptors (Ref. 36). Here, we specifically reduced Ppc's to assess whether the electron flow from Ppc's to OmcS nanowires could be occur in this direction.

OmcS nanowire charging via...

This is a nice bioinformatics section and is interesting and suggestive of the more widespread importance of this work.

We are very grateful for these appreciative comments by the reviewer.

Finally, – the periplasm is not a space, but this term is used a few times. It is a subcellular compartment. In one section the authors say it is a 'space' and then say it is a 'crowded' environment (lines 44-47)! A 'space' can't be crowded so I prefer to call it the periplasmic compartment. Actually, some 35 years ago it was shown it likely has a texture of thick porridge –it is indeed a crowded environment.

As suggested, we have rephrased this as periplasmic compartment.

Reviewer #2 (Remarks to the Author)

Report on manuscript no. NCOMMS-23-36005, entitled “A widespread extracellular electron transfer pathway for charging microbial 1 cytochrome OmcS nanowires via periplasmic cytochromes PpcABCDE” by P.C. Portela *et al.*

The authors have taken on a challenging chemical problem in a biological system that has widespread interest for those studying electron transfer in Nature. The manuscript is relatively densely written, although there is a lot of ground to cover to ensure the reliability of the interpretations, which do appear plausible and correct.

We are very grateful for these appreciative comments by the reviewer.

Some statements are unclear. However, some clarifications and reordering of material might make the ultimate paper more accessible to the reader. Some suggestions and some other questions arising from the manuscript are listed below. If the authors respond satisfactorily to these then I will be pleased to recommend this work for publication in *Nature Communications*.

We have revised the manuscript as per all suggestions to ensure that there are no unclear statements.

- 1) **P 1, l 37, refer to Figure 1(b) at the first mention of the nanowires to make this report more accessible to readers, and place this figure closer to its first call from the text.**

As suggested, we have moved the figure closer to the text and referred the fig. 1(b).

- 2) **P 2, l 52, reference to another multi-haem species that has had a reduction potential measured in both (near) native and isolated environments would support this highly likely assertion.**

We have referred reduction potential of solvent-exposed hemes in microperoxidases (Ref. 31) because we are unaware of studies comparing the reduction potential of native vs. isolated multi-heme cytochromes.

- 3) **P 2, l 82, reference 8 does not explicitly mention the Cotton effect, but appears to use changes in CD spectra and comparisons with calculations using density functional theory to draw conclusions about their nanowires. Some further clarification on this point might aid the reader.**

As suggested, we have now described the Cotton effect in the revised manuscript and also added a new reference 8 that describes in detail how it is a hallmark of excitonic coupling.

- 4) **P 2, Figure 1(b), here and elsewhere there is no mention of inter-haem geometry and any requirements of that to facilitate electron transfer.**

We have now revised the text to describe this figure and inter-haem geometry and any requirements of that to facilitate electron transfer as follows: Structures of these nanowires reveal interconnected chains of cytochromes encasing stacked heme cofactors arranged in parallel (3.4-4.1 Å) and T-stacked (5.4-6.1 Å) sequential pairs (Fig. 1b and Ref. 11). Such closely stacked hemes can promote rapid and insulated electron conduction over distances of several micrometers.

- 5) **P 3, l 85, the statement about potentials lying within tens of millivolts is not supported by the content of Figure 2(d), and there appears to be a 1-2-2-1 pattern of grouping of potentials.**

As suggested by the reviewer, we have removed the text regarding this to avoid confusion.

- 6) **P 3, l 93, it is unclear if there was any microscopy performed (if it were possible) on the nanowires that were adsorbed on gold to ascertain location and orientation.**

We have discussed in the revised manuscript that the atomic force microscopy and infrared nanospectroscopy imaging have shown that OmcS nanowires adsorbed on gold show overall secondary structure similar to cryo-EM structure and cover the entire surface without any specific orientation.

- 7) **P 3, ll 110-11, 50 kD and 10 kDa and the 1,000-fold multiplier are unclear.**

We apologize for being unclear. We have rewritten this section as follows: The OmcS monomer is ca. 5 nm long and has molecular weight of ~50 kDa. Therefore, a nanowire of typical length of 1 micrometer contains at least 200 monomers with total molecular weight 10,000 kDa (50 kDa x 200). The molecular weights of PpcA-E are ~10 kDa, which are 1000-fold lower than OmcS nanowires, and, consequently, the NMR signals of PpcA-E narrow and easily distinguishable in contrast to those from OmcS nanowire.

8) P 4, Figure 3, labelling is difficult to read.

As suggested, we have increased the font size for the labeling to make it more readable.

9) It is unclear how to compare the voltages in Figure 2(d), in Supplementary Table 1 and in Figure 5

We have clarified that all potentials are midpoint reduction potentials of hemes vs. SHE. Figure 5 is a graphical depiction of midpoint reduction potential values mentioned in Supplementary Table 1. The voltages in Figure 2(d) concern the proposed midpoint reduction potential values and Figure 5 are comparable and Figure 4

10) P 4, l 117, “As OmcS monomer contains six hemes and each PpcA-E contain 3 hemes,” – the compositions do not seem to have been set out clearly early on in the manuscript and are confusing at this point in the development of the science. This also affects understanding the argument on line 123 about the heme ratio.

We thank the reviewer for pointing this out. We have stated these compositions in the beginning of the manuscript when we first introduce OmcS and PpcA-E.

11) P 4, Figure 3, caption, the oxidized form should be light blue and not blue?

We have made this change.

12) P 5, l 146, “the chemical shifts were within the reported for labile complexes, an essential characteristic in electron transfer reactions.” – meaning is unclear.

We have clarified this as follows: “The variation in chemical shifts ($\Delta\delta$) indicates whether the interaction between two proteins involves a well-defined interaction surface or it is highly dynamic and there are many conformations that the partner proteins adopt. We observe small variation in the chemical shifts ($\Delta\delta < 0.15$ ppm) in comparison with other cytochrome complexes (Ref. 41). Therefore, Ppc-OmcS interaction is labile, which is essential in electron transfer reactions. Notably, $\Delta\delta_{\text{PpcC}}$ (0.08-0.14 ppm) was significantly higher than $\Delta\delta_{\text{PpcABCDE}}$ (< 0.04 ppm) (Fig. 3c), indicating a stronger heme interaction of PpcC with OmcS, likely due to larger binding interface. This strongest heme interaction explains the highest electron transfer of PpcC among paralogs, despite its lowest periplasmic abundance³⁶.

13) P 5, caption to Figure 4, this is confusing: “with units of kcal/e• mol” – what is this?

We apologize for this error. The units have been corrected to kT/e at $T=298$ K.

14) “Combined chemical shift” – what is this? “highest $\Delta\delta_{\text{combined}}$ ” how is this defined?

We have probed the most affected backbone signals of PpcA in its interaction with OmcS: as such, we have chemical shift variation values for ^1H and ^{15}N nuclei. To integrate this data into one single value, we calculated the combined chemical shift value ($\Delta\delta_{\text{combined}}$) of ^1H and ^{15}N nuclei according to the procedure indicated in the Materials and Methods section which is based on the approach by Schumann and co-workers (Ref. 42) and that is widely used by the NMR community. To clarify this point, we have added in the manuscript a remark that a more detailed description of this procedure is indicated in Methods. The cut-off value is the calculation of the standard deviation of the $\Delta\delta_{\text{combined}}$ values, according to the procedure proposed by Schumann and co-workers and that is widely adopted in the NMR community (Ref. 42). The “highest $\Delta\delta_{\text{combined}}$ ” is defined as the residues which present a $\Delta\delta_{\text{combined}} > 1.5\Delta\delta_{\text{cut-off}}$.

15) P 6, l 161, “the most affected residues” – how has this been quantified, what criteria were used?

As suggested by the reviewer, we have discussed this in greater detail in the revised manuscript. This question is related to the previous one: the quantification of the most affected backbone signals was performed using the established NMR procedure of calculating a cut-off value based on the standard deviation of the set of $\Delta\delta_{\text{combined}}$ values determined in this work (see Materials and Methods). The most affected residues were the ones that presented $\Delta\delta_{\text{combined}} > 1.5\Delta\delta_{\text{cut-off}}$. We have clarified this aspect in the manuscript, along with the clarifications provided in the previous question.

16) P 6, l 166, “An 82 mV- positive”, what does this mean?

We have clarified that this term means 82 mV more positive redox potential.

- 17) **P 6, caption to Figure 5, “determined from 1-99% reduction/oxidation.” – what does this mean?**
The redox window is calculated as the voltage range from when 1% of the protein is reduced to when 99% of it is reduced. We used “reduced/oxidized” because it is the same definition for reduction or oxidation. We have now referred to this as “reduction” for simplicity.
- 18) **P 6, caption to Figure 5, “suggests the stacking of surface-exposed hemes for efficient electron injection” –what does this mean?**
We apologize for the confusion. We meant that the Fig. 5b inset suggests that surface-exposed hemes of PpcA and OmcS can come close to each other during interaction. We have reworded the legends by changing the work “stacking” to “proximity” of surface-exposed hemes.
- 19) **P 7, ll 209-10, “This contrasts with the long-standing model that EET requires periplasmic cytochromes exchanging electrons among each other.” In the new model presented here, is this exchange still possible, and from what has now been revealed about binding is there a consequence for cytochromes residing in the periplasm?**
We have clarified this section as follows. Our studies eliminate the need for slow diffusion of electrons among monomeric periplasmic cytochromes by direct binding to nanowires. Due to this direct binding, periplasmic cytochromes will not cause the bottleneck for electron transfer.
- 20) **P 8, there seems to be some repetition in the text around here, the very neat new results are in danger of being obscured. Note also that a “Third” point is made twice (lines 247 and 248).**
We have removed all the repetitive text in the revised manuscript and removed the “third” point.
- 21) **P 8, l 271, “PpcA and a porin-cytochrome complex can be sufficient for EET to soluble iron, which is reduced on the bacterial surface and thus only limited to short range.” – what does this mean?**
We have rewritten this text as follows: Prior genetic studies have shown that deleting most abundant genes encoding outer-surface and periplasmic cytochromes, except PpcA and a OmcB porin-cytochrome complex, still enables EET to Fe(III) citrate. But this is a soluble iron, which is reduced on the bacterial surface and thus only limited to short range electron transfer.
- 22) **P 11, caption, “larges” ?**
We have corrected the typo to “largest”
- 23) **Pages 12-13, tidy up the presentation of some of the experimental work: quantities_space_units**
As suggested, we have corrected them in the revised manuscript.
- 24) **P14, l 424, page numbers 1 – 100 ?**
We have added page numbers to this reference.

Reviewer #3 (Remarks to the Author)

This is a great manuscript with important information about the electron transfer mechanisms in *Geobacter sulfurreducens*. Learning that periplasmic cytochromes are able to transfer electrons to a cytochrome nanowire opens up possibilities that simplify the current view of how EET occurs.

We are very grateful for these appreciative comments by the reviewer.

It also opens up a series of questions that, while not the main purpose of the manuscript, would be great for the authors to discuss. We have added the discussion in the revised manuscript as per the reviewer’s suggestions.

I only have a few comments and suggestions on discussion topics that would help improve the manuscript.

- 1) **My first thought when reading the title was: “what about the Ext cytochromes?”. These are not only thought to be crucial components of the outer membrane, but in this paper from the Bond lab (10.1128/JB.00347-18) were shown to be needed for metal reduction or anode electron delivery. If nanowires can directly accept electrons from the periplasm, why are these outer membrane**

cytochromes so crucial in EET? I do not expect the authors to answer this question, but the lack of discussion of these cytochromes and the current view on how electron transfer takes place (vs. a view where nanowires take electrons from PpcABCDE) is largely missing.

As suggested by the reviewers, we have discussed the role of Ext cytochromes as follows: Besides outer-surface nanowires, *G. sulfurreducens* also relies on five porin-cytochrome complexes on the outer-membrane, among which ExtABCD are essential in bacterial growth only on electrodes, which does not require OmcS nanowires^{25,26},

- 2) **There is an embedded assumption that OmcS nanowires start at the periplasm in order to interact with PpcABCDE, then crossing the OM. I know this is a theory previously proposed by the Malvankar lab, starting with the studies on PilA. Nonetheless, the importance of this is not mentioned here. As far as I know this is still a theory (no specific evidence of nanowires in the cytochrome periplasmic space), but an important one to discuss in the context of the findings of the manuscript.** As suggested by the reviewers, we have discussed in the manuscript this as follows: As OmcS remains in periplasm when they are not fully assembled into nanowires¹⁰, our model assumes that nanowire formation initiates in the periplasm itself (Fig. 5b).
- 3) **Many publications have shown reduction potentials of *Geobacter sulfurreducens* biofilms, most of which have a midpoint around -150 mV. The low potentials previously reported for OmcS was conflicting with the knowledge of biofilm electrochemistry. In this sense, the new potential reported for OmcS clears up a lot of discrepancies in the field (e.g., expression studies, deletion studies suggesting omcS is associated to a -150mV midpoint potential pathway). I suggest the authors point this out as an important contribution of their work.** As suggested, we have pointed out how the identification of correct redox potential for OmcS nanowires reconciles prior discrepancies and suggest a simple model for EET pathway as follows: *G. sulfurreducens*' biofilm redox response is centered at -150 mV, highlighting the central role of the periplasmic cytochrome family in the electron transfer chain.²⁰ However, at the time of the study, the reduction potential value of OmcS (-212 mV) suggested that the electron transfer to OmcS was not thermodynamically favorable²⁴. Our spectroelectrochemical analysis on intact OmcS nanowires in this work showed an 82 mV more positive reduction potential with a 40 mV-larger redox-active window than that reported previously. The larger overlap in potentials indicates OmcS could receive electrons from PpcA-E, suggesting a new EET pathway with minimal components that could provide a more direct pathway for extracellular electron transfer (Fig. 5a,b).
- 4) **Page 1 ln 47 – I would not dismiss diffusion as “slow” ($10^5/s$) based on calculations of electron transfer between cytochromes (Reference 19). Diffusion is concentration dependent, so a 10x increase in electron rates can easily be achieved by having 10x periplasmic cytochromes to nanowires. In any case, the discussion on redox potentials after this is much more compelling to set the stage for your manuscript.** As suggested, we have included the further discussion of redox potential in the revised manuscript. We have clarified that to overcome the difference in diffusion, periplasmic cytochromes need to be at least 10-times higher concentration than nanowires. As OmcS nanowires typically contain 200 monomers, this will require at least more than 2000 monomers of periplasmic cytochromes to overcome the diffusion limitation assuming electron transfer rate is similar among OmcS and periplasmic cytochromes.
- 5) **Page 3 ln 99 – While I understood the term, I found it inadequate to call the electron transfer by PpcC “efficient” because it was able to relatively transfer more electrons than other Ppc cytochromes. Especially when considering that the higher electron transfer is probably associated to a higher difference in redox potentials, the term efficiency seems weird in this context.**

We apologize for the confusion and have now clarified this term further as follows: Surprisingly, PpcC showed a slight oxidation even in the 1:1 heme ratio, indicating higher electron transfer to OmcS than the other periplasmic cytochromes. This efficient oxidation is due to higher binding to OmcS, as quantified by NMR chemical shift perturbation experiments (Fig. 3c). We have clarified this in the revised manuscript.

REVIEWERS' COMMENTS

Reviewer #2 (Remarks to the Author):

The authors have made a thorough, careful and complete response to the comments of the reviewers of their manuscript, which I am pleased to recommend for publication in Nature Communications.

Reviewer #3 (Remarks to the Author):

I do not have any further comments. The authors have addressed my comments.